# Effect of Light of Different Spectral Compositions on Pro/Antioxidant Status, Content of Some Pigments and Secondary Metabolites and Expression of Related Genes in Scots Pine

**DOI:** 10.3390/plants12132552

**Published:** 2023-07-05

**Authors:** Pavel Pashkovskiy, Yury Ivanov, Alexandra Ivanova, Alexander Kartashov, Ilya Zlobin, Valery Lyubimov, Aleksandr Ashikhmin, Maksim Bolshakov, Vladimir Kreslavski, Vladimir Kuznetsov, Suleyman I. Allakhverdiev

**Affiliations:** 1K.A. Timiryazev Institute of Plant Physiology, Russian Academy of Sciences, Botanicheskaya Street 35, 127276 Moscow, Russia; pashkovskiy.pavel@gmail.com (P.P.); ivanovinfo@mail.ru (Y.I.); aicheremisina@mail.ru (A.I.); botanius@yandex.ru (A.K.); ilya.zlobin.90@mail.ru (I.Z.); vlkuzn@mail.ru (V.K.); 2Institute of Basic Biological Problems, Russian Academy of Sciences, Institutskaya Street 2, 142290 Pushchino, Russia; valyulyub@gmail.com (V.L.); ashikhminaa@gmail.com (A.A.); lfbv22@gmail.com (M.B.); vkreslav@rambler.ru (V.K.)

**Keywords:** antioxidants, hydrogen peroxide, *Pinus sylvestris*, light spectral composition, carotenoids, secondary metabolites, gene expression

## Abstract

The aim of this study was to investigate the effect of light quality (white fluorescent light, WFL, containing UV components), red light (RL, 660 nm), blue light (BL, 450 nm), and white LED light (WL, 450 + 580 nm) on the components of the cellular antioxidant system in *Pinus sylvestris* L. in needles, roots, and hypocotyls, focusing on the accumulation of key secondary metabolites and the expression of related genes. The qualitative and quantitative composition of carotenoids; the content of the main photosynthetic pigments, phenolic compounds, flavonoids (catechins, proanthocyanidins, anthocyanins), ascorbate, and glutathione; the activity of the main antioxidant enzymes; the content of hydrogen peroxide; and the intensity of lipid peroxidation (MDA and 4-HNE contents) were determined. RL resulted in an increase in the content of hydrogen peroxide and 4-HNE, as well as the total fraction of flavonoids in the needles. It also enhanced the expression of several *PR* (pathogen-related) genes compared to BL and WL. WFL increased the content of phenols, including flavonoids, and enhanced the overall activity of low-molecular antioxidants in needles and hypocotyls. BL increased the content of ascorbate and glutathione, including reduced glutathione, in the needles and simultaneously decreased the activity of peroxidases. Thus, by modifying the light quality, it is possible to regulate the accumulation of secondary metabolites in pine roots and needles, thereby influencing their resistance to various biotic and abiotic stressors.

## 1. Introduction

Varying the light spectral composition is one of the strategies to accelerate the growth of conifers under artificial conditions, which is crucial for the development of technologies aimed at producing sustainable seedlings required to preserve existing forest areas [1]. Therefore, studying the effects of different compositions of light on plant health and development is important.

Light intensity and quality are key environmental factors for plant growth and development [2,3]. Notably, alterations in light quality can influence the pigment content and pigment composition, as well as the physiological, morphological, and biochemical characteristics of plants [3,4,5]. At the beginning of their lifecycle, many plants grow under the shade of taller species, which changes the quality of light they receive [6].

On the one hand, red light (RL, 620–700 nm) and blue light (BL, 400–500 nm) in the visible region of the radiation spectrum are the most effective for photosynthesis and thus for plant growth. On the other hand, blue light, perceived by cryptochromes and phototropins, is involved in many processes of photobiosynthesis and photomorphogenesis, in maintaining circadian rhythms, and in the functioning of stomata [7]. There has been noticeable progress in understanding the effects of RL and BL on flowering herbaceous plants [8]; however, the responses of conifers have not yet been fully explored. Conifer populations are widespread and can, as a result, adapt to lighting with significant qualitative and quantitative differences [9]. RL, detected by phytochromes, controls a wide range of plant responses. These include pigment biosynthesis, seed germination, hypocotyl development, and the adaptation of plants to various stress factors. Phytochromes participate in the different responses of plants, including pine seedlings, affecting growth, plant morphology, and the activity of crucial enzymes [10].

It is likely that under RL exposure, phytochromes play a crucial role in these processes. For instance, the content of photosynthetic pigments and photosystem II (PSII) activity were measured in chloroplasts of 14-day-old *Pinus sylvestris* L. seedlings grown under light and dark conditions [11]. The RL pre-irradiation of dark-grown seedlings induced pigment accumulation, especially chlorophyll *b* accumulation, and activated chloroplast activity. However, the application of far-red light (FRL) after RL exposure inhibited pigment content and PSII activity. The authors concluded that the accumulation of pigments and PSII activity in pine seedlings are controlled by phytochrome.

Additionally, BL and FRL participate in the control of *P. sylvestris* growth and photosynthetic processes [12,13]. Thus, data obtained in the study [13] showed that BL and the deficiency of FRL limited the growth of Scots pine seedlings and that northern populations are more sensitive to BL than southern populations.

It was demonstrated that the physiological and morphological responses of shade-intolerant Scots pine seedlings to light quality were more pronounced than those of shade-tolerant Norway spruce seedlings [12]. Growth under the FR-containing light treatments produced tall seedlings with larger needle dry mass and Pn. The removal of FR in the light reduced the height growth and altered the branching patterns in the seedlings but did not affect the Chl content. The growth and Pn value in Scots pine seedlings depend on FRL addition, that is, the R/FR ratio acting by phytochrome on pine seedling growth and photosynthesis [12,13]. However, there are few data on the effects of light quality on secondary metabolites, especially different pigments, in pine seedlings.

BL and FRL also play a role in controlling growth and photosynthetic processes [12,13]. BL and FRL deficiency limited the growth of Scots pine seedlings. Northern populations were found to be more sensitive to BL than southern populations. Moreover, the results suggested that one of the additional BL photoreceptors, zeitlupe (ZTL), and the transcription factor HY5 are involved in light-induced growth regulation of seedlings and that CRY2 plays a role in shade avoidance [13].

It has been established that the synthesis of certain secondary metabolites is light-dependent and regulated by photoreceptors and various components associated with light signaling, including transcription factors. Some secondary metabolites have antioxidant properties and enhance the plant’s adaptive potential to abiotic stressors and biopathogens [14,15]. The spectral composition of light not only regulates morphogenetic and photosynthetic reactions in plants but also modifies their pro/antioxidant balance. However, there is scant information on how light quality influences the content of secondary metabolites and the pro/antioxidant status of woody plants, especially conifers.

The aim of our study was to explore how light of different spectral compositions could affect the pro/antioxidant status and secondary metabolites of Scots pine seedlings. We studied the relationship between photomorphogenesis processes and the expression of genes essential for the biosynthesis of various secondary metabolites. Additionally, we determined which pigments, secondary metabolites, and antioxidant enzymes contribute the most to the antioxidant system of pine plants under illumination with different spectral compositions.

## 2. Results

### 2.1. The Content of Hydrogen Peroxide and Products of Lipid Peroxidation

All types of LEDs led to a significant increase in the content of hydrogen peroxide in the needles of seedlings: from 2.3 times on WL to 3.8 times on RL (Table 1). Against this background, an increase in the content of lipid peroxidation products in needles was noted only on RL (the content of 4-HNE was 54.5% higher than that under WFL) and on WL (the MDA content was 18.3% higher than that under WFL) (Table 1). It is noteworthy that on BL, despite the increase in the content of hydrogen peroxide, the content of 4-HNE decreased by 31.2% compared to that on WFL (Table 1).

A significant effect of lighting conditions on the content of hydrogen peroxide in the roots of seedlings was noted, except for a decrease in its content by 2.5 times under BL. At the same time, no differences in the content of lipid peroxidation products in the roots of seedlings were found (Table 1).

### 2.2. Activity of Antioxidant Enzymes

Different LED light led to a slight decrease in superoxide dismutase (SOD) activity in needles, most pronounced on RL (−8.6%) and BL (−14.3%) compared to WFL (Table 2). Changes in SOD activity were noted in the roots. An increase (+30.8%) was noted in WL, and a decrease (−28.9%) in SOD activity was noted in RL compared to WFL (Table 2).

On BL, there was a reduced level of guaiacol peroxidase activity in both roots (−29.6%) and needles (−40.8%) compared to WFL. For other variants of the experiment, no changes in the activity of guaiacol peroxidase were noted (Table 2).

Only BL led to a decrease in the activity of ascorbate peroxidase in the needles of seedlings by 20.9% compared to WFL. In all other variants, the level of ascorbate peroxidase activity in roots and needles was maintained at a level comparable to that in WFL (Table 2).

No effect of light on the activity of catalase in the organs of seedlings was found (Table 2).

### 2.3. The Content of Glutathione and Ascorbic Acid

In roots, the glutathione content was very low and therefore it could not be reliably determined [16,17]. In particular, the GSSG content exceeded the total GSH content, which made it impossible to calculate the reduced GSH content (Table 3).

Only BL led to changes in the content of glutathiones in the needles of seedlings, so the content of total GSH increased by an average of 32.1%, GSH by 30.7%, and GSSG by 2.2 times in comparison with other types of light. The roots of seedlings were also characterized by an increase in the content of total GSH under BL by 1.9 times in comparison with WFL but without changes in the content of GSSG (Table 3).

Under BL in the needles of seedlings, the content of ascorbic acid increased by 36.0%, and under WL and RL, its content remained at a level comparable to that under WFL (Table 3).

### 2.4. TEAC and Content of Total Phenolics, Flavonoids (Catechins, Proanthocyanidins, Anthocyanins), and Lignin

Narrow band light led to a decrease in the antioxidant capacity of low molecular weight antioxidants (TEAC) in all variants relative to WFL in needles and seedling hypocotyls by an average of 24.3% and 20.3%, respectively. TEAC in seedling roots did not differ from WFL when exposed to LEDs (Table 4).

The needles and hypocotyls of seedlings on WFL were characterized, on average, by 1.4 and 1.3 times higher contents of phenolic compounds than on other types of light. The content of total flavonoids in the needles and hypocotyls of seedlings also reached a maximum at WFL and was similar to the content of phenolic compounds (Table 4). It is noteworthy that the content of flavonoids in the roots and hypocotyls exceeded their content in the needles by at least five times. At the same time, no significant differences in the content of phenols and flavonoids in the roots of seedlings were found (Table 4).

In the WL and BL variants, a reduced content of catechins and proanthocyanidins in the needles of seedlings was noted in comparison with WFL, on average by 37.4% and 35.8%, respectively. It is noteworthy that against the background of the minimum content of catechins and proanthocyanidins in the needles on BL, their content in the root was maximum, exceeding WFL by 24.2% and 10.5%, respectively (Table 4). In all variants of LEDs, the content of catechins and proanthocyanidins in seedling hypocotyls was lower than that in WFL hypocotyls. RL had the lowest content of catechins (−33.9%) and proanthocyanidins (−26.3%) compared to WFL (Table 4).

A statistically significant change in the content of anthocyanins was noted only on BL, where their level was lower than that on WFL by 21.8% (Table 4).

The minimum content of lignin in the organs of seedlings was observed under the action of BL and was most pronounced (−19.5% compared to WFL) in the roots of seedlings. Under the influence of WL, an increase of 11.3% in the content of lignin in the hypocotyls of seedlings was observed (Table 4) and the highest content of lignin in the needles was observed on the RL.

### 2.5. Content of Carotenoids

The impact of LEDs was accompanied by qualitative changes in the composition of carotenoids represented by eight compounds. Lutein, β-carotene, α-carotene, and violaxanthin accounted for approximately 90% of the total pool of carotenoids in all variants of the experiment. Common features of the effect of LEDs were a decrease in the content of α-carotene (on average, 2-fold), β-carotene, and lutein compared to WFL (Figure 1). A characteristic feature of the impact of BL and RL was an increase in the content of the unknown carotene by more than three times, in comparison with WFL (Figure 1). The influences of BL and RL manifested itself in the content of antheraxanthin, the level of which on BL increased compared to that on WFL, and on RL, it decreased below the detection limits (Figure 1). The content of zeaxanthin decreased by three-fold in WL and by four-fold in BL compared with WFL (Figure 1).

### 2.6. Gene Expression

Against the background of RL, a two-fold increase in the level of *4CL* (4-coumarate-CoA ligase) gene transcripts was observed in needles, while the intensity of this gene expression was maximum in roots (Figure 2). When plants were irradiated with RL in needles, the *LAR* gene also increased by more than 1.5 times, while the level of *LAR* gene transcripts in the roots was 1.5 times higher against the background of BL rather than RL (Figure 2). The level of transcripts of genes for the biosynthesis of salicylic acid *PR1* and *PR5* in needles increased by RL by more than seven and three times, respectively, while in the roots, this trend persisted, but the expression of the *PR5* gene increased by five times relative to the WFL control (Figure 2). At the same time, the transcript levels of the jasmonate biosynthesis genes *JAZa*, *JAZb*, and *MYC* in RL increased by more than two times relative to all other light variants (Figure 2).

## 3. Discussion

In previous work, we compared the effects of white light (WL), white fluorescent light (WFL), BL, and RL on the growth of Scots pine plantlets. We found that RL or BL exposure resulted in a larger biomass of *P. sylvestris* plants compared to WL and WFL [6]. The content of photosynthetic pigments also peaked under RL. Our previous work on Scots pine plantlets demonstrated that growth was optimal under WFL and RL, and the accumulation of photosynthetic pigments was also maximal under RL exposure compared to WL or BL [17].

It was proposed that optimal growth might be due to a substantial accumulation of carbon in the needles. This corresponds to a balanced interaction between photosynthesis and respiration, as well as the increased activity of cytokinins and auxins in seedlings. Although *P. sylvestris* seedlings can grow under low RL at the start of ontogenesis, improved RL radiation can significantly enhance their growth and development.

Our experiments showed a dependence of the activity of antioxidant enzymes on the quality of the light. For instance, under WFL, we observed a reduced content of hydrogen peroxide in needles, accompanied by an increase in SOD activity (Table 2). At the same time, on BL, the content of lipid peroxidation products (such as 4-HNE content) was reduced (Table 1). On BL, we observed the minimum activity of guaiacol-dependent peroxidase in the roots and needles and the lowest activity of ascorbate peroxidase in the needles (Table 2). However, the maximum content of ascorbic acid in the needles and oxidized and reduced glutathione in the needles and roots was observed (Table 3). We hypothesize that BL has a selective positive effect on the antioxidant system via the accumulation of ascorbate, one of the most potent antioxidants, and glutathione, but not through the activation of antioxidant enzymes (Table 2). In contrast, RL had practically no effect on the level of glutathione in the needles and roots of seedlings. It appears that the induction of ascorbate and reduced glutathione is one of the possible pathways of the acclimation of pine seedlings to BL.

Secondary metabolites typically have antioxidant activity, which is critically important for plant adaptation to changing environmental conditions [18]. Our data suggest that the highest activity of low-molecular weight antioxidants was observed in the needles and in the hypocotyl under WFL (Table 4). Moreover, the content of phenolic compounds, flavonoids, catechins, and proanthocyanidins changed similarly, suggesting a significant contribution of these groups of secondary metabolites to the total antioxidant activity (Table 4). We propose that this might be due to the presence of UV light in the WFL spectrum, which induces the formation of secondary metabolites (Table 4). RL also had a stimulating effect on the accumulation of catechins, flavonoids, and proanthocyanidins (Table 4). In roots, the maximum level of Trolox equivalent antioxidant capacity (TEAC) and various phenolic compounds was under BL (Table 4). It appears that the effect of BL and RL on TEAC and the content of various phenolic compounds are facilitated by the LAR enzyme. The maximum transcript levels of the *LAR* gene were observed in the needles under RL and in the roots under BL (Figure 2). The lack of interconnection between TEAC and the content of various phenolic compounds, on the one hand, and the *LAR* transcription level encoding proanthocyanidin biosynthesis, on the other hand, can be explained through the negative effect of UV on the expression of this gene. However, *PAL* gene expression can be stimulated by UV (Figure 2). The changes in phenolic and flavonoid content under different light conditions in the experiment are in line with data of Agati et al. 2013 who noted that exposure to different light qualities led to significant changes in flavonoid content in plant tissues [19]. This is consistent with our findings where maximum flavonoid content was observed under white fluorescent light (WFL), which provides a broad spectrum (Table 4). The differences in catechin and proanthocyanidin content between light conditions may be associated with the plant’s adaptation mechanism to different light qualities, as these compounds are potent antioxidants and are associated with photoprotection. A study by Yang et al. 2018 explored how changes in light quality and quantity lead to adjustments in these secondary metabolite levels [20]. The lowered anthocyanin and flavonoid content under blue light (BL) is intriguing, considering that many previous studies have found an increased production of anthocyanins under BL [21,22]. Differences between species and varietal differences of pine might explain these discrepancies [23]. Lignin content is known to be responsive to various environmental factors, including light [24], and it has previously been suggested that alterations in light conditions can modulate lignin biosynthesis, which might explain the observed changes under different LED variants. For example, the reduced lignin content under BL and its increased content under RL in needles (Table 4) are likely due to a bigger accumulation of carbon under RL than under BL. The active form of phytochrome significantly increases under RL conditions, which, through G-protein, calmodulin, and cGMP pathways, activates the expression of associated genes. This, in turn, induces oxidative lignin deposition by regulating the diversion of photosynthetic carbon sources towards the lignin synthesis pathway [25,26]. At the same time, there does not necessarily have to be a positive correlation between the formation of wood and the lignin content in trees [27]. These results suggest a complex interaction between light quality and secondary metabolite production. We also assume that in pine plants, flavonoids and secondary metabolites contribute more to the antioxidant system than the enzymatic system, which is also confirmed by our results (Table 2, Table 3 and Table 4).

Overall, it is worth noting the specific effect of RL on the expression of jasmonate and salicylic acid genes (Figure 2). Although jasmonates and salicylic acid are secondary metabolites, their gene products are involved in hormonal signaling, which seems to be associated with red light signaling in pine seedlings. Salicylic acid plays a major role in the plant defense response, and its biosynthesis is influenced by light conditions, as demonstrated by the increase in *PR1* and *PR5* gene transcripts upon RL exposure, which is consistent with previous research [28]. Jasmonic acid is a critical hormone involved in plant responses to stress and signal transduction, and its regulation by RL has been explored by Robson et al. 2010 [29], which is consistent with our results (Figure 2).

Carotenoids are not only one of the main photosynthetic pigments but also secondary metabolites with pronounced antioxidant properties [30]. It is known that carotenoids absorb light from UV-A and blue and green regions, and their activity can be increased when plants are exposed to BL [31]. We showed that the highest content of total carotenoids and xanthophylls was observed under WFL (Figure 1). Thus, WFL and BL most effectively influence the content of carotenoids that are important for protecting photosystem II from photooxidation. Furthermore, the content of antheraxanthin decreased with RL and increased with BL (Figure 1). Additionally, antheraxanthin, which, along with zeaxanthin, is formed in the xanthophyll cycle from the thylakoid-associated pigment violaxanthin [32], can be involved in a stress-protective mechanism [33]. Therefore, under BL and WFL, a change in the carotenoid composition may be one of the protective mechanisms against unfavorable conditions. The differences in the content of antheraxanthin, which decreased under RL and increased under BL, may be due to the increased content of hydrogen peroxide in the tissues under BL and especially under RL (Table 1). Although there is a doubling in the amount of hydrogen peroxide compared to BL, its content is still within the normal physiological range. This likely ties more to a general metabolic activation than to a heightened state of oxidative stress. This assumption is further supported by the minor fluctuations in the activity of low-molecular weight antioxidants and minimal variations in the lipid peroxidation levels (Table 1).

Pathogenesis-related proteins (PRs) are specifically induced in response to infection by pathogens such as fungi, bacteria, and viruses or to adverse environmental factors through a defense mechanism associated with the increased production of reactive oxygen species (ROS) in the plant cell [34]. The transcription level of the genes encoding these proteins increased significantly only under RL exposure. We also observed elevated levels of H_2_O_2_ after RL exposure. This implies that only RL, but not BL, can enhance the induced resistance of pine seedlings, likely through H_2_O_2_ induction. This light must constitute a significant fraction of the visible spectrum for the growth of such seedlings under artificial light conditions. This is also in line with the high transcriptional level of the *MYC* gene in needles of seedlings grown under RL, which participates in jasmonic acid signaling and is involved in plant-induced resistance, as well as through the alteration of pro-/antioxidant balance into the induction of biosynthesis of secondary metabolites [35]. This is in agreement with the enhanced content of H_2_O_2_ in plants grown under RL conditions compared to WL and BL [6,18].

## 4. Materials and Methods

### 4.1. Plant Materials and Experimental Design

Seeds of Scots pine (*Pinus sylvestris* L.) were provided by the Training and Experimental Forestry Enterprise of the Bryansk State Technological University of Engineering (Bryansk, Russia) and were collected in the Bryansk region from high-productivity pine stands in complex forest types. Seeds were germinated in a hydroculture on individual substrates in polypropylene cartridges filled with 1% agar bungs in individual boxes of the climatic chamber under red LEDs (RL, maxima of 660 nm), blue LEDs (BL, maxima of 450 nm), white LED light (WL, maxima of 450 and 580 nm), and white fluorescent lamps (WFL, 58 W/33–640, white fluorescent lamps (Philips, Pila, Poland), 130 ± 10 µmol (photons) m^–2^ s^–1^ containing a 2.7 µmol (photons) m^–2^ s^–1^ UV-B component with maximum 311 nm) for 6 weeks [17]. The light intensity was equal across all light variants. The spectral characteristics of the light sources were determined using an AvaSpec-ULS2048CL-EVO spectrometer (Avantes B.V. Oude Apeldoornseweg, Apeldoorn, The Netherlands) (Appendix A). After seed coat rupture and cotyledon expansion, the seedlings were transferred to a nutrient solution [36]. The seedlings were cultivated in 6 L plastic trays (171 seed beds per tray) in a growth chamber that provided a constant air temperature of 24 ± 2 °C and a 16 h photoperiod. The nutrient solutions were constantly aerated and renewed once a week. During the week, a constant volume of the nutrient solution was maintained by adding distilled water [17].

Six-week-old plants were collected, and after washing the roots with distilled water and blotting them on filter paper, 1–3 seedlings were grouped together and dissected into their organs. Samples were fixed in liquid nitrogen and stored at −70 °C until the biochemical analyses.

### 4.2. Determination of H_2_O_2_

To determine the content of H_2_O_2_, frozen samples (50–100 mg) of needles or roots were transferred to 0.4 mL of 2 M trichloroacetic acid (TCA) and homogenized. The homogenate was triply flushed into a test tube using 1 mL of 0.05 M potassium–phosphate buffer (pH 7.0). Next, 3.5 mL of the homogenate was mixed with 100 mg of activated coal for the sorption of pheophytin and carotenoids and centrifuged for 20 min at 10,000× *g*. The supernatant was decanted and titrated with 2 M KOH to pH 7.0. The content of H_2_O_2_ in 100 µL of the extract was determined by measuring bioluminescence in a mixture of 10^−6^ M horseradish peroxidase and 10^−4^ M luminol (1:1, total volume 1 mL and expressed in µmol H_2_O_2_ g^−1^ FW) [37].

### 4.3. Lipid Peroxidation

The contents of malondialdehyde (MDA) and 4-hydroxyalkenals were determined spectrophotometrically, with a maximum optical absorption at 586 nm, by measuring the product that formed during the reaction using a selective reagent, 1-methyl-2-phenylindole (Aldrich, CAS Number 3558-24-5, St. Louis, MO, USA), in accordance with Gérard-Monnier et al. (1998). 1,1,3,3-Tetraethoxypropane was used to construct the calibration curve [36,38].

### 4.4. Enzyme Activities

A weighed sample of the plant material of approximately 200 mg was ground in a porcelain mortar in liquid nitrogen and extracted using 2 mL of ice-cold 100 mM carbonate–bicarbonate buffer (pH = 10.3) containing 200 mg of suspended insoluble poly(vinylpolypyrrolidone) (Sigma-Aldrich, Burlington, MA, USA, CAS Number 9003-39-8), 10 mM DL-dithiothreitol (Sigma-Aldrich, Burlington, MA, USA, D0632), and 5 mM phenylmethyl sulfonyl fluoride (PMSF) (Sigma-Aldrich, Burlington, MA, USA, 329-98-6). The homogenate was centrifuged at 15,000× *g* for 10 min, and the supernatant obtained was used for the determination of enzyme activities.

Catalase was assayed by measuring the initial rate of the disappearance of hydrogen peroxide by the method of Chance and Maehly [39].

The guaiacol peroxidase activity was determined by the method proposed by Ridge and Osborne [40]. The dynamics of the changes in optical density were recorded on a spectrophotometer for 3 min at a wavelength of 470 nm. Measurements were taken every 2 s.

The ascorbate peroxidase activity was determined spectrophotometrically according to the rate of destruction of ascorbic acid by the method of Nakano and Asada [41].

The activity of superoxide dismutase (SOD). To remove low-molecular-weight compounds that could interfere with the definition of SOD activity, the crude extract was passed through “illustra NAP-5 Columns” (GE Healthcare Life Sciences, 17-0853-02, Singapore), according to the manufacturer’s instructions. All steps in the preparation of the enzyme extract were carried out at 5 °C. An aliquot of the extract was used to determine its protein content. Total SOD activity was measured by the percentage of reaction inhibition rate of the enzyme with WST-1 substrate (a water-soluble tetrazolium dye) and xanthine oxidase using an SOD Assay Kit (Sigma-Aldrich, Burlington, MA, USA, 19160), according to the manufacturer’s instructions. Each endpoint assay was monitored by absorbance at 450 nm (the absorbance wavelength for the colored product of the WST-1 reaction with superoxide) after 20 min of reaction time at 37 °C. The percentage of inhibition was normalized per milligram of protein and presented as SOD activity units [38].

The protein content in the preparations was determined using a Bicinchoninic Acid Protein Assay Kit (Sigma-Aldrich, Burlington, MA, USA, B9643), according to the manufacturer’s instructions.

### 4.5. Ascorbic Acids

Needles (0.1 g) or roots (0.2 g) were ground in liquid nitrogen and extracted with 3% (*w*/*v*) metaphosphoric acid plus 10 mM EDTA for 20 min at +4 °C. Supernatant obtained via centrifugation extract at 13,000× *g* for 20 min at +4 °C was used to ascorbic acid assay by Ascorbic Acid Assay Kit (K-ASCO, Megazyme Ltd., Wicklow, Ireland), according to the manufacturer’s protocol. The concentration of ascorbic acid was calculated using a calibration curve.

### 4.6. Glutathione Content

The total and oxidized glutathione contents were determined in frozen plant samples, as described by Rahman et al. [16] with the following modifications: the quantity of triethanolamine added to the reaction medium was decreased by two-fold to keep the pH below 7.0 and prevent glutathione oxidation. The total and oxidized glutathione contents were expressed in nmol g^−1^ DW [42].

### 4.7. Anthocyanin Extraction and Measurement

Anthocyanin was extracted and determined spectrophotometrically [43]. Briefly, 0.10–0.15 g of needles was ground in liquid nitrogen and incubated in 600 μL of extraction buffer (methanol containing 1% HCl) in an ultrasonic bath for 15 min and overnight at 4 °C in the dark. After extraction, samples were centrifuged for 5 min at 10,000× *g*, the supernatant was transferred into a 2 mL centrifuge tube, 400 μL of water and 400 μL of chloroform were added to each sample and vortexed, followed by centrifugation at 10,000× *g* and 4 °C for 5 min, and the absorbance of the supernatant was read at 530 (A530) and 657 (A657) nm. The anthocyanin content was calculated as follows: (A530 − 0.33 × A657) g^−1^ fresh weight. Each sample was extracted and measured in three independent experiments.

### 4.8. HPLC Analysis of Carotenoids

Carotenoid analysis was performed as described previously [44,45]. The HPLC device (Shimadzu, Kyoto, Japan) consisted of (1) an LC-10ADVP pump with an FCV-10ALVP module, (2) a detector with an SPD-M20A diode matrix, and (3) a CTO-20 AC thermostat. The separation of the carotenoids was performed on a 4.6 × 250 mm reversed-phase column (Agilent Zorbax SB-C18, Agilent, Santa Clara, CA, USA) at 22 °C. The carotenoids were identified by their retention time and absorption spectra. The quantification of each carotenoid was performed by comparing its peak area in the region of 270–800 nm to the sum of all carotenoid peaks taken as 100% and was calculated with the LC-solution program (Shimadzu, Kyoto, Japan) using the molar extinction coefficients described elsewhere [46].

### 4.9. Phenolic Compound Contents

Phenolic compounds were extracted with 80% methanol from samples ground in liquid nitrogen.

The low-molecular-weight antioxidant capacity (Trolox equivalent antioxidant capacity (TEAC)) was determined spectrophotometrically, according to the method described by Re et al. [47], involving the reaction of methanolic extracts with 2,2’-azino-bis[3-ethylbenzothiazoline-6-sulfonic acid] diammonium salt (ABTS) (Sigma-Aldrich, Burlington, MA, USA, CAS number 30931-67-0).

The total phenolics were determined spectrophotometrically using Folin and Ciocalteu’s phenol reagent (Sigma-Aldrich, Burlington, MA, USA, MDL number MFCD00132625), according to the procedure described by Singleton and Rossi [48]. The total phenolic content was expressed as gallic acid equivalents (GAE) in milligrams per gram of fresh weight (FW).

The total flavonoids were measured according to the methods of Kim et al. [49]. Afterwards, 1000 µL of distilled water, 150 µL of extracted sample, and 50 µL of 5% NaNO_2_ were mixed together. After 6 min, 50 µL of 10% AlCl_3_ was added, and after another 5 min, 300 µL of 1 M NaOH was added to the mixture. The reaction mixture was homogenized, and after 10 min, the absorbance at 510 nm was measured. The total flavonoids were calculated by constructing a calibration curve using (+)-catechin hydrate (Sigma-Aldrich, Burlington, MA, USA, CAS Number 225937-10-0) and were expressed as milligrams of (+)-catechin per gram of FW [43].

The total content of catechins and proanthocyanidins (PA) was determined spectrophotometrically through the reaction of catechins, PA, and 1% vanillin in acidic media [50]. The catechin and PA contents were calculated by constructing a calibration curve using (+)-catechin hydrate and were expressed as milligrams of (+)-catechin per gram of FW [51].

Additionally, the PA content was determined via reaction with butanol reagent. The butanol reagent was prepared by mixing 128 mg of FeSO_4_ 7H_2_O and 5 mL of HCl together, and the reaction was completed in 100 mL of n-butanol. A total of 50 μL of extracted sample and 700 μL of butanol reagent were mixed together, after which the mixture was heated at 95 °C for 45 min [52]. The sample was cooled, and the absorbance at 550 nm was measured. The total PA content was calculated by constructing a calibration curve using cyanidin chloride (PhyProof^®^, PHL80022) and was expressed as cyanidin equivalents in milligrams per gram of FW [51].

### 4.10. Cell-Wall Preparation and Lignin Quantification

Cell-wall preparation was performed according to the method of Lange et al. [53] with minor modifications. The samples (approximately 250 mg) of frozen roots, hypocotyls and needles were ground in a mortar in the presence of liquid nitrogen. The resulting fine powder was suspended in 1.5 mL of methanol (Sigma-Aldrich, Burlington, MA, USA, CAS 67-56-1) and transferred to 2 mL centrifuge tubes. The mixture was vigorously stirred for 1 h and centrifuged (12,000× *g*, 5 min). The pellet was consecutively treated with 1.5 mL of the following solvents and solutions with mixing for 15 min, followed by centrifugation for 5 min as described above: (a) methanol (twice), (b) 1 M NaCl, (c) 1% (*w*/*v*) sodium dodecyl sulfate (Sigma-Aldrich Burlington, MA, USA, CAS 151-21-3), (d) H_2_O (twice), (e) ethanol, (f) chloroform/methanol (l:l, *v*/*v*), and (g) tert-butyl methyl ether (Sigma-Aldrich, Burlington, MA, USA, CAS 1634-04-4). The remaining insoluble material (purified cell walls) was freeze-dried (Labconco FreeZone 2.5 L Benchtop Freeze Dry System, Kansas City, MO, USA) overnight.

Lignin was assayed via derivatization with thioglycolic acid (Sigma-Aldrich, Burlington, MA, USA, CAS 68-11-1) according to the method of Lange et al. [53]. Approximately 15 to 20 mg of the purified cell-wall preparations were placed in a 1.5 mL screw-cap tube and treated with 1 mL of 2 M HC1 and 0.2 mL of thioglycolic acid for 4 h at 95 °C. After cooling to room temperature, the mixture was centrifuged for 20 min at 20,000× *g*. The supernatant was removed, and the remaining pellet was washed three times with H_2_O. The pellet was suspended in 1 mL of 0.5 M NaOH and vigorously shaken overnight to extract the LTGA. Following centrifugation as described above, the supernatant was transferred into a 2 mL centrifuge tube, and the pellet was washed with 0.5 mL of 0.5 M NaOH. The combined alkali extract was acidified with 0.3 mL of concentrated HCl, and the LTGA was allowed to precipitate at 4 °C for 4 h. The mixture was centrifuged as described above, the supernatant was removed, and the brown pellets were dried in a Vacufuge plus (Eppendorf, Germany). The pellet was dissolved in 1 mL of 0.5 M NaOH.

The absorbance of the samples was measured with a Genesys 10 UV–Vis spectrophotometer (Thermo Fisher Scientific, Waltham, MA, USA) at a wavelength of 280 nm. A calibration curve was obtained using alkali lignin (Sigma-Aldrich, Burlington, MA, USA, CAS Number 8068-05-1).

### 4.11. RNA Extraction and RT-PCR

RNA isolation was performed according to the method of Kolosova et al. [54] modified by Pashkovskiy et al. [55]. The quantity and quality of the total RNA were determined using a NanoDrop 2000 spectrophotometer (Thermo Fisher Scientific, Waltham, MA, USA). cDNA synthesis was performed using the M-MLV Reverse Transcriptase Kit (Fermentas, Waltham, MA, USA) and the oligo (dT) 21 primer. The expression patterns of the genes were assessed using the CFX96 Touch™ Real-Time PCR Detection System (Bio-Rad, Hercules, CA, USA). Gene-specific primers (Appendix A) for cinnamate 4-hydroxylase (*CH4*, A0A023 W7L9), 4-coumarate-CoA ligase (*4CL*, B0ZQ68), leucoanthocyanidin reductase (*LAR*, BN000697.1), phenylalanine ammonia-lyase (*PAL,* Q8RUZ3), phytoene synthase (*PSY*, MA 407452g0010), pathogen-related 1 (*PR1*, EF084624.1), pathogen-related 5 (*PR5*, MZ222278.1), Jasmonate-Zim domain 1 (*JAZa*, EF083399.1), Jasmonate-Zim domain 2 (*JAZb*, GILO01669683.1), and bHLH transcription factors (*MYC*, MF395340.1) were selected using nucleotide sequences from the National Center for Biotechnology Information (NCBI) database (www.ncbi.nlm.nih.gov (accessed on 1 December 2021); UniProt.org (accessed on 1 December 2021); congenie.org (accessed on 1 December 2021)) with Vector NTI Suite 9 software (Invitrogen, Waltham, MA USA). The transcript levels were normalized to the expression of the *Actin1* gene. The experiments were performed with 6 biological and 3 analytical replicates. The relative gene expression signal intensity in WFL plants was considered 1.

### 4.12. Statistics

Each plant sample fixed in liquid nitrogen was treated as a biological replicate. Therefore, there were 10 biological replicates for the determination of enzyme activities; 6 biological replicates for lipid peroxidation products (i.e., glutathione, anthocyanin, and phenolic compounds), lignin contents, and gene expression analyses; and 3 biological replicates for the HPLC analysis of carotenoids.

The data were statistically analyzed using SigmaPlot 12.3 (Systat Software, San Jose, CA, USA) with one-way analysis of variance (ANOVA) followed by Duncan’s method for normally distributed data (in the figures, significant differences are denoted by different normal letters) and the Kruskal-Wallis one-way ANOVA on ranks followed by the Student–Newman-Keuls post hoc test for nonnormally distributed data and data with unequal variance (in the figures, significant differences are denoted by different italic letters for the Student–Newman-Keuls post hoc test). Different letters were used to indicate significance at *p* ≤ 0.05.

Pairwise comparisons of the means were performed between WFL and other variants using Student’s *t*-test for normally distributed data or the Mann–Whitney rank sum test when the *t*-test was not applicable. The values presented in the tables and figures are the arithmetic means ± standard errors.

## 5. Conclusions

Scots pine seedlings most actively responded to the action of RL. This was evidenced by an increased content of hydrogen peroxide and the more intense expression of pathogenesis-related and jasmonic acid signaling genes. The stress-protective effect of RL is achieved through the increased accumulation of secondary metabolites in the needles. The stress-protective effect of WFL seems to be associated with the increased formation of secondary metabolites, including some carotenoids, due to the presence of UV-B in the spectrum. BL caused an increase in the content of ascorbate and glutathione, including reduced forms, in the needles. Another feature of the effect of BL on pine seedlings was the stimulation of the expression of the *PAL* and *LAR* genes in the roots, which resulted in an increase in the flavonoid content. Thus, by manipulating the quality of light, it is possible to alter the biosynthesis of secondary metabolites in the roots and needles of pine seedlings within certain limits, thereby regulating their resistance to various biotic and abiotic stressors.

## Figures and Tables

**Figure 1 plants-12-02552-f001:**
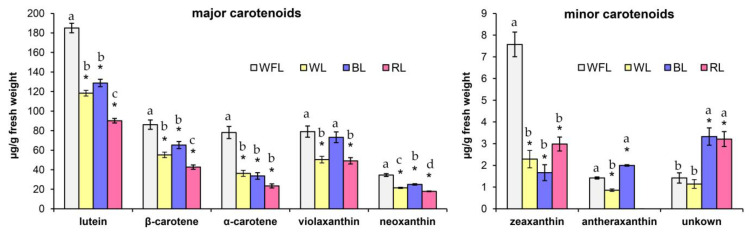
The carotenoid compositions in µg g^−1^ FW under the influence of red LEDs (RL, maxima of 660 nm), blue LEDs (BL, maxima of 450 nm), white LED light (WL, maxima of 450 and 580 nm), and white fluorescent lamps (WFL, white fluorescent lamps) after 6 weeks of the experiment. Asterisks indicate significant differences (*p* ≤ 0.05) between the experimental treatments for each carotenoid compared to WFL (*t*-test). Different letters indicate significant differences (*p* ≤ 0.05) between the experimental treatments. The means ± standard errors, *n* = 3.

**Figure 2 plants-12-02552-f002:**
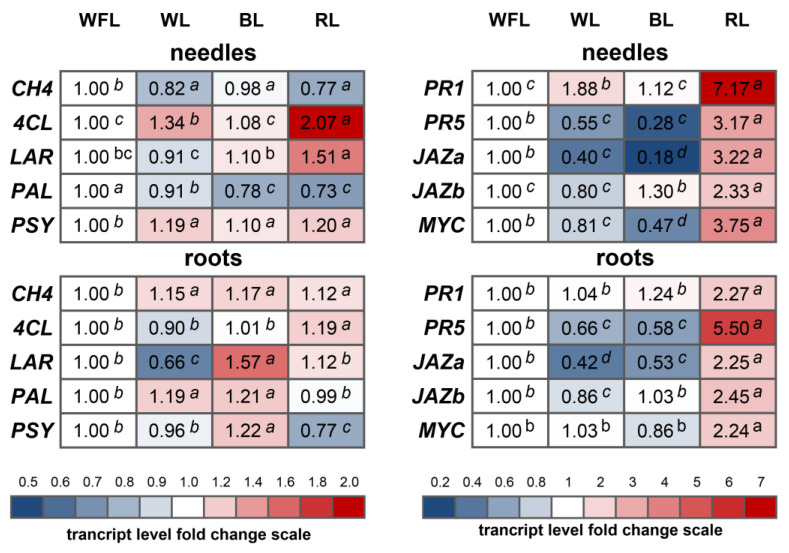
Transcript levels of cinnamate 4-hydroxylase (*CH4*), 4-coumarate-CoA ligase (*4CL*), leucoanthocyanidin reductase (*LAR*), phenylalanine ammonia-lyase (*PAL*), phytoene synthase (*PSY*), pathogen-related 1 (*PR1*), pathogen-related 5 (*PR5*), Jasmonate-Zim domain 1 (*JAZa*), Jasmonate-Zim domain 2 (*JAZb*), and bHLH transcription factors (*MYC*) under red LEDs (RL, maxima of 660 nm), blue LEDs (BL, maxima of 450 nm), white LED light (WL, maxima of 450 and 580 nm), and white fluorescent lamps (WFL) after 6 weeks of the experiment. The transcript levels were normalized to the expression of the *Actin1* gene. The gene expression at WFL was used as one unit. Different italic letters indicate significant differences (*p* ≤ 0.05) between the experimental treatments for each gene. *n* = 6.

**Table 1 plants-12-02552-t001:** Effect of light of different spectral compositions on the contents of hydrogen peroxide, MDA and 4-HNE.

	White Fluorescent Light (WFL)	White Light (WL)	Blue Light (BL)	Red Light (RL)
roots				
H_2_O_2_, µmol g^−1^ FW	0.139 ± 0.009 *a*	0.218 ± 0.062 *a*	0.056 ± 0.014 *b*,×	0.117 ± 0.006 *a*
MDA, nmol g^−1^ FW	8.42 ± 0.61 a	7.40 ± 0.56 a	8.72 ± 0.30 a	9.08 ± 1.00 a
4-HNE, nmol g^−1^ FW	27.92 ± 3.60 a	24.38 ± 1.11 a	25.57 ± 3.10 a	30.68 ± 2.91 a
needles				
H_2_O_2_, µmol g^−1^ FW	0.054 ± 0.004 c	0.125 ± 0.006 b,*	0.142 ± 0.008 b,*	0.207 ± 0.009 a,×
MDA, nmol g^−1^ FW	11.35 ± 0.47 b	13.43 ± 0.48 a,*	11.23 ± 0.60 b	10.15 ± 0.55 b
4-HNE, nmol g^−1^ FW	159.4 ± 17.0 b	169.4 ± 9.3 b	109.6 ± 7.3 c,*	246.3 ± 12.8 a,*

Different letters within each row indicate significant differences (*p* ≤ 0.05), according to ANOVA on ranks, followed by Duncan’s method (regular letters) or Kruskal–Wallis one-way ANOVA on ranks, followed by the Student–Newman-Keuls post hoc test (italic letters). Pairwise comparisons of the means were performed between WFL and other variants using Student’s *t*-test for normally distributed data (significant differences at *p* ≤ 0.05 denoted by asterisk (*)) or the Mann–Whitney rank sum test when the *t*-test was not applicable (significant differences at *p* ≤ 0.05 denoted by multiplication symbols (×)). MDA = malondialdehyde; 4-HNE = 4-hydroxyalkenals.

**Table 2 plants-12-02552-t002:** Effect of light of different spectral compositions on antioxidant enzyme activity.

	White Fluorescent Light (WFL)	White Light (WL)	Blue Light (BL)	Red Light (RL)
Roots				
POD, µmol guaiacol min^−1^ mg^−1^ protein	78.72 ± 8.40 *a*	74.47 ± 7.79 *a*	55.43 ± 7.93 *a*,*	72.91 ± 10.06 *a*
SOD, relative units mg^−1^ protein	103.8 ± 10.0 *b*	135.8 ± 13.4 *a*,×	103.6 ± 10.8 *b*	73.76 ± 5.66 *c*,*
APX, µmol ascorbic acid min^−1^ mg^−1^ protein	139.6 ± 13.5 a	161.6 ± 12.4 a	138.1 ± 11.9 a	143.1 ± 15.8 a
CAT, mmol H_2_O_2_ min^−1^ mg^−1^ protein	7.11 ± 0.69 *a*	8.63 ± 0.61 *a*	6.90 ± 0.64 *a*	7.29 ± 0.81 *a*
Needles				
POD, µmol guaiacol min^−1^ mg^−1^ protein	17.82 ± 1.07 *a*	18.19 ± 1.10 *a*	10.55 ± 0.78 *b*,*	15.68 ± 1.82 *a*
SOD, relative units mg^−1^ protein	101.4 ± 2.3 a	96.25 ± 2.05 ab	86.94 ± 2.49 b,*	92.65 ± 1.25 b,×
APX, µmol ascorbic acid min^−1^ mg^−1^ protein	63.50 ± 5.04 a	64.21 ± 2.98 a	50.22 ± 3.69 a,*	56.36 ± 4.41 a
CAT, mmol H_2_O_2_ min^−1^ mg^−1^ protein	12.91 ± 1.14 a	15.53 ± 1.52 a	12.28 ± 0.97 a	13.23 ± 1.55 a

Different letters within each row indicate significant differences (*p* ≤ 0.05) according to ANOVA on ranks, followed by Duncan’s method (regular letters) or the Kruskal-Wallis one-way ANOVA on ranks, followed by the Student–Newman-Keuls post hoc test (italic letters). Pairwise comparisons of the means were performed between WFL and other variants using Student’s *t*-test for normally distributed data (significant differences at *p* ≤ 0.05 denoted by asterisk (*)) or the Mann–Whitney rank sum test when the *t*-test was not applicable (significant differences at *p* ≤ 0.05 denoted by multiplication symbols (×)). POD = guaiacol peroxidase; SOD = superoxide dismutase; APX = ascorbate peroxidase; CAT = catalase.

**Table 3 plants-12-02552-t003:** Effect of light of different spectral compositions on the contents of ascorbic acid and glutathione.

	White Fluorescent Light (WFL)	White Light (WL)	Blue Light (BL)	Red Light (RL)
Roots				
GSH total, nmol g^−1^ FW	1.57 ± 0.34 b	2.25 ± 0.40 ab	3.03 ± 0.27 a,*	0.87 ± 0.31 b
GSSG, nmol g^−1^ FW	1.33 ± 0.16 a	1.36 ± 0.35 a	1.64 ± 0.31 a	1.61 ± 0.20 a
Needles				
Ascorbic acid, µmol g^−1^ FW	2.11 ± 0.21 a	2.19 ± 0.17 a	2.87 ± 0.22 a	2.34 ± 0.23 a
GSH total, nmol g^−1^ FW	407.1 ± 17.7 b	395.7 ± 27.3 b	515.1 ± 15.9 a,*	368.8 ± 30.8 b
GSH, nmol g^−1^ FW	400.0 ± 18.0 b	388.6 ± 26.6 b	501.3 ± 18.2 a,*	363.6 ± 30.8 b
GSSG, nmol g^−1^ FW	3.55 ± 0.37 b	3.59 ± 1.12 b	6.91 ± 1.36 a,*	2.58 ± 0.31 b

Different letters within each row indicate significant differences (*p* ≤ 0.05) according to ANOVA on ranks, followed by Duncan’s method (regular letters). Pairwise comparisons of the means were performed between WFL and other variants using Student’s *t*-test (significant differences at *p* ≤ 0.05 denoted by asterisk (*)). GSH total = total glutathione; GSH = reduced glutathione; GSSG = oxidized glutathione.

**Table 4 plants-12-02552-t004:** Effect of light of different spectral compositions on antioxidant capacity and phenolic compound content.

	White Fluorescent Light (WFL)	White Light (WL)	Blue Light (BL)	Red Light (RL)
Roots				
TEAC, µmol Trolox g^−1^ FW	43.87 ± 1.64 a	43.30 ± 1.36 a	47.87 ± 2.09 a	44.43 ± 1.60 a
GAE, mg g^−1^ FW	3.05 ± 0.11 ab	2.80 ± 0.08 b	3.33 ± 0.14 a	3.22 ± 0.12 a
Flavonoids, mg catechin g^−1^ FW	2.91 ± 0.11 ab	2.64 ± 0.07 b	3.20 ± 0.14 a	3.01 ± 0.09 a
Catechins + proanthocyanidins, mg catechin g^−1^ FW	2.96 ± 0.15 b	2.87 ± 0.09 b	3.68 ± 0.23 a,*	3.42 ± 0.18 ab
Proanthocyanidins, mg cyanidin g^−1^ FW	1.26 ± 0.06 ab	1.16 ± 0.03 b	1.39 ± 0.06 a	1.34 ± 0.06 a
Lignin, mg g^−1^ DW	83.66 ± 3.59 a	83.78 ± 2.42 a	67.38 ± 2.88 b,*	80.44 ± 3.43 a
Hypocotyls				
TEAC, µmol Trolox g^−1^ FW	87.57 ± 2.27 a	70.32 ± 2.17 b,*	65.83 ± 2.15 b,*	73.13 ± 4.18 b,*
GAE, mg g^−1^ FW	4.44 ± 0.13 a	3.70 ± 0.14 b,*	3.45 ± 0.11 bc,*	3.25 ± 0.18 c,*
Flavonoids, mg catechin g^−1^ FW	3.49 ± 0.10 a	2.77 ± 0.10 b,*	2.59 ± 0.09 b,*	2.45 ± 0.16 b,*
Catechins + proanthocyanidins, mg catechin g^−1^ FW	5.34 ± 0.26 a	4.69 ± 0.18 ab	4.31 ± 0.24 b,*	3.54 ± 0.27 c,*
Proanthocyanidins, mg cyanidin g^−1^ FW	2.07 ± 0.07 a	1.74 ± 0.07 b,*	1.67 ± 0.06 b,*	1.53 ± 0.09 b,*
Lignin, mg g^−1^ DW	138.6 ± 2.2 *b*	154.3 ± 3.0 *a*,*	132.1 ± 2.5 *c*	144.5 ± 4.3 *b*
Needles				
TEAC, µmol Trolox g^−1^ FW	49.28 ± 2.65 a	35.93 ± 1.44 b,*	36.50 ± 1.85 b,*	39.50 ± 2.26 b,*
GAE, mg g^−1^ FW	2.16 ± 0.10 a	1.46 ± 0.13 b,*	1.62 ± 0.08 b,*	1.58 ± 0.12 b,*
Flavonoids, mg catechin g^−1^ FW	0.75 ± 0.04 a	0.53 ± 0.03 bc,*	0.48 ± 0.03 c,*	0.60 ± 0.03 b,*
Catechins + proanthocyanidins, mg catechin g^−1^ FW	1.67 ± 0.12 a	1.18 ± 0.10 bc,*	0.92 ± 0.11 c,*	1.30 ± 0.13 b
Proanthocyanidins, mg cyanidin g^−1^ FW	0.65 ± 0.04 a	0.46 ± 0.03 bc,*	0.37 ± 0.03 c,*	0.51 ± 0.05 b
Anthocyanins, relative level mg^−1^ FW	0.156 ± 0.012 a	0.127 ± 0.014 a	0.122 ± 0.008 a,*	0.167 ± 0.020 a
Lignin, mg g^−1^ DW	29.88 ± 1.07 ab	31.11 ± 1.11 ab	27.74 ± 1.03 b	32.79 ± 1.43 a

Different letters within each row indicate significant differences (*p* ≤ 0.05) according to ANOVA on ranks, followed by Duncan’s method (regular letters) or the Kruskal-Wallis one-way ANOVA on ranks, followed by the Student–Newman-Keuls post hoc test (italic letters). Pairwise comparisons of the means were performed between WFL and other variants using Student’s *t*-test (significant differences at *p* ≤ 0.05 denoted by asterisk (*). TEAC = Trolox equivalent antioxidant capacity; GAE = gallic acid equivalents.

## Data Availability

The datasets generated and/or analyzed during the current study are available from the corresponding author on reasonable request.

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
