# Peer review of "Effect of Light of Different Spectral Compositions on Pro/Antioxidant Status, Content of Some Pigments and Secondary Metabolites and Expression of Related Genes in Scots Pine"

_plants, 2023, doi:10.3390/plants12132552_

Round 1

Reviewer 1 Report

General comments

The article is interesting; however, there are some parts that need to be improved. The discussion part is a bit poor. It should be improved by including more international literature.

Specific comments:

Results:

2.3. The content of glutathione and ascorbic acid

Lines 154-158. The analysis method used by Rahman et al. [17] did not allow us to reliably deter-155 mine the glutathione content in Scots pine roots.

Are there other methods of doing this?

Why have they not been used?

2.5. Content of carotenoids

Figure 1. It is recommended to replace asterisks with letters to indicate significant differences.

Discussion:

Many results are presented but there is little discussion. Their explanation needs to be improved. It is recommended to improve the discussion with citations from international scientific publications.

Author Response

1. The article is interesting; however, there are some parts that need to be improved. The discussion part is a bit poor. It should be improved by including more international literature.

Answer: We improved the discussion.

We have added new text to the manuscript:

«The changes in phenolic and flavonoid content under different light conditions in the experiment are in line with data of Agati et al. 2013 who noted that exposure to different light qualities led to significant changes in flavonoid content in plant tissues [19]. This is consistent with our findings where maximum flavonoid content was observed under white fluorescent light (WFL), which provides a broad spectrum (Table 4). The differences in catechin and proanthocyanidin content between light conditions may be associated with the plant's adaptation mechanism to different light qualities, as these compounds are potent antioxidants and are associated with photoprotection. A study by Yang et al. 2018 explored how changes in light quality and quantity lead to adjustments in these secondary metabolite levels [20]. The lowered anthocyanin and flavonoid content under blue light (BL) is intriguing, considering that many previous studies have found an increased production of anthocyanins under BL [21,22]. Differences between species and varietal differences of pine might explain these discrepancies [23]. Lignin content is known to be responsive to various environmental factors, including light [24], and it has previously been suggested that alterations in light conditions can modulate lignin biosynthesis, which might explain the observed changes under different LED variants. For example, the reduced lignin content under BL and its increased content under RL in needles (Table 4) are likely due to a bigger accumulation of carbon under RL than under BL. The active form of phytochrome significantly increases under RL conditions, which, through G-protein, calmodulin, and cGMP pathways, activates the expression of associated genes. This, in turn, induces oxidative lignin deposition by regulating the diversion of photosynthetic carbon sources towards the lignin synthesis pathway [25,26]. At the same time, there does not necessarily have to be a positive correlation between the formation of wood and the lignin content in trees [27]. These results suggest a complex interaction between light quality and secondary metabolite production. We also assume that in pine plants, flavonoids and secondary metabolites contribute more to the antioxidant system than the enzymatic system, which is also confirmed by our results (Tables 2-4).

Overall, it is worth noting the specific effect of RL on the expression of jasmonate and salicylic acid genes (Fig. 2). Although jasmonates and salicylic acid are secondary metabolites, their gene products are involved in hormonal signaling, which seems to be associated with red light signaling in pine seedlings. Salicylic acid plays a major role in the plant defense response, and its biosynthesis is influenced by light conditions, as demonstrated by the increase in PR1 and PR5 gene transcripts upon RL exposure, which is consistent with previous research [28]. Jasmonic acid is a critical hormone involved in plant responses to stress and signal transduction, and its regulation by RL has been explored by Robson et. al. 2010 [29], which is consistent with our results (Fig. 2).»

2. Lines 154-158. The analysis method used by Rahman et al. [17] did not allow us to reliably deter-155 mine the glutathione content in Scots pine roots.

Are there other methods of doing this?

Why have they not been used?

Answer: The formulation used in the previous version of the Manuscript was quite confusing. In fact, the glutathione content in roots was very low and therefore unsuitable to measure. In the revised version of the Manuscript, this sentence was changed to the following: ‘In roots, the glutathione content was very low and therefore it could not be reliably determined’’.

3. Figure 1. It is recommended to replace asterisks with letters to indicate significant differences.

Answer: It is done.

4. Discussion:

Many results are presented but there is little discussion. Their explanation needs to be improved. It is recommended to improve the discussion with citations from international scientific publications.

Answer: We improved the discussion.

Reviewer 2 Report

Plants can use light not only as a source of energy for photosynthesis, but also as information for adapting to environmental conditions. Using a network of photoreceptors, plants can fine-tune their metabolism in response to the spectral composition of light. Over the past decade, significant progress has been made in understanding the effects of light spectral composition on angiosperm growth and development. However, the light response of gymnosperms has not yet been fully explored. In the manuscript by Pashkovskiy et al, the authors show how light of different spectral compositions affects the pro/antioxidant status and secondary metabolites of Scots pine seedlings. The authors provide the interesting results, but some points need to be clarified.

 Minor points:

 1)    In the Abstract authors declare that “the aim of this study was to investigate the effect of light quality (white fluorescent light, WFL, containing UV-component), red light (RL, 660 nm), blue light (BL, 450 nm), and white LED light (WL, 450 + 580 nm)”. However, the Methods state that WL light had maxima at 660 and 580 nm (Line 351).

2)    It is unclear why the authors refer to the combination of 450 and 580 nm as white light. Shouldn't white light include the entire PAR spectrum (blue, green, red)?

3)    Why did the authors avoid combining blue and red light?

4)    Line 353: The spectral characteristics of the light sources were determined using an AvaSpec-ULS2048CL-EVO spectrometer (Figure S1). But Fig. S1 is missing.

5)    Please indicate whether the light intensity (130±10 μmol (photons) m-2 s-1) refers to all light sources used (Line 352).

6)    For WFL lump, please specify the wavelength of the UV component.

7)    The authors conclude that there was “more intense expression of pathogenesis-related proteins" (Line 561). However, they examined the expression of genes encoding the PR-proteins but not the expression of proteins.

8)    Authors conclude that “by manipulating the quality of light, it is possible to alter the biosynthesis of secondary metabolites in the roots and needles of pine seedlings within certain limits, thereby regulating their resistance to various biotic and abiotic stressors”. Based on the data obtained, can the authors recommend the most optimal light regime for the growth of resistant pine seedlings?

Author Response

1. In the Abstract authors declare that “the aim of this study was to investigate the effect of light quality (white fluorescent light, WFL, containing UV-component), red light (RL, 660 nm), blue light (BL, 450 nm), and white LED light (WL, 450 + 580 nm)”. However, the Methods state that WL light had maxima at 660 and 580 nm (Line 351).

Answer: 450 + 580 nm is correct, we improve the text.

2. It is unclear why the authors refer to the combination of 450 and 580 nm as white light. Shouldn't white light include the entire PAR spectrum (blue, green, red)?

Answer: Thank you for your question. Indeed, we indicated the peak values that the spectrometer shows for these LEDs. The white LEDs utilized were broad-spectrum, producing blue, red, and green light, but primarily peaking at 450nm and 580nm. These LEDs emit a warm white light. The light source specification is given in the supplementary.

3. Why did the authors avoid combining blue and red light?

Answer: In our study, we specifically investigated how red and blue light independently affect plant growth. Other light sources we used served as controls, encompassing a broad range of wavelengths. In our future research, we intend to explore not only the combined impact of red and blue light on plants, but also the effect of different red-to-blue light ratios.

4. Line 353: The spectral characteristics of the light sources were determined using an AvaSpec-ULS2048CL-EVO spectrometer (Figure S1).But Fig. S1 is missing.

Answer: We are grateful to the reviewer for pointing out the inaccuracy, we have added a figure S1.

5. Please indicate whether the light intensity (130±10 μmol (photons) m-2s-1) refers to all light sources used (Line 352).

Answer: We add the phrase: «The light intensity was equal across all light variants.»

6. For WFL lump, please specify the wavelength of the UV component.

Answer: We add the phrase to Materials and metgods section: « …containing 2.7 µmol (photons) m –2 s –1 UV-B component 311 nm.». The light source specification is given in the supplementary.

7. The authors conclude that there was“more intense expression of pathogenesis-related proteins" (Line 561). However, they examined the expression of genes encoding the PR-proteins but not the expression of proteins.

Answer: We improve the text.

8. Authors conclude that “by manipulating the quality of light, it is possible to alter the biosynthesis of secondary metabolites in the roots and needles of pine seedlings within certain limits, thereby regulating their resistance to various biotic and abiotic stressors”. Based on the data obtained, can the authors recommend the most optimal light regime for the growth of resistant pine seedlings?

Answer: Despite the conducted studies and our prior research, we cannot definitively ascertain what type of light is necessary for all conifers. However, we can unambiguously affirm that any light containing as much red spectrum as possible will be more beneficial compared to other light options for Pinus sylvestris. Additionally, we plan to incorporate red light and UV-B radiation in our future studies.

Round 2

Reviewer 1 Report

The proposed changes have been made